

# Ant mosaics in Bornean primary rain forest high canopy depend on spatial scale, time of day, and sampling method

Kalsum M. Yusah[1], William A. Foster[2], Glen Reynolds[3] and Tom M. Fayle[4,5]

[1] Institute for Tropical Biology and Conservation, Universiti Malaysia Sabah, Kota Kinabalu, Sabah, Malaysia
[2] Department of Zoology, University of Cambridge, Cambridge, Cambridgeshire, United Kingdom
[3] The SE Asia Rainforest Research Partnership (SEARRP), Lahad Datu, Sabah, Malaysia
[4] Biology Centre of the Czech Academy of Sciences, Institute of Entomology, Ceske Budejovice, South Bohemia, Czech Republic
[5] Faculty of Science, University of South Bohemia, Ceske Budejovice, South Bohemia, Czech Republic

Corresponding author
Kalsum M. Yusah,
kalsum.myusah@gmail.com

## ABSTRACT

**Background**. Competitive interactions in biological communities can be thought of as giving rise to "assembly rules" that dictate the species that are able to co-exist. Ant communities in tropical canopies often display a particular pattern, an "ant mosaic", in which competition between dominant ant species results in a patchwork of mutually exclusive territories. Although ant mosaics have been well-documented in plantation landscapes, their presence in pristine tropical forests remained contentious until recently. Here we assess presence of ant mosaics in a hitherto under-investigated forest stratum, the emergent trees of the high canopy in primary tropical rain forest, and explore how the strength of any ant mosaics is affected by spatial scale, time of day, and sampling method.

**Methods**. To test whether these factors might impact the detection of ant mosaics in pristine habitats, we sampled ant communities from emergent trees, which rise above the highest canopy layers in lowland dipterocarp rain forests in North Borneo (38.8–60.2 m), using both baiting and insecticide fogging. Critically, we restricted sampling to only the canopy of each focal tree. For baiting, we carried out sampling during both the day and the night. We used null models of species co-occurrence to assess patterns of segregation at within-tree and between-tree scales.

**Results**. The numerically dominant ant species on the emergent trees sampled formed a diverse community, with differences in the identity of dominant species between times of day and sampling methods. Between trees, we found patterns of ant species segregation consistent with the existence of ant mosaics using both methods. Within trees, fogged ants were segregated, while baited ants were segregated only at night.

**Discussion**. We conclude that ant mosaics are present within the emergent trees of the high canopy of tropical rain forest in Malaysian Borneo, and that sampling technique, spatial scale, and time of day interact to determine observed patterns of segregation. Restricting sampling to only emergent trees reveals segregatory patterns not observed in ground-based studies, confirming previous observations of stronger segregation with increasing height in the canopy.

## INTRODUCTION

Species' colonisation of natural habitats can be influenced by various factors, including interactions with other species and variation in the physical environment. Understanding the assembly rules that describe these processes is a fundamental objective of ecology (*Gotelli & McCabe, 2002*). Inter-specific competition, a process that is well known to influence species assembly, should be greatest between species that have similar niches, for example in terms of their morphology, function and pattern of resource use. This produces distinctive patterns of species co-occurrence (*Gotelli & Ellison, 2002*). Species that share the same resource should co-occur less often than expected by chance, and conversely, where species do co-occur, they should differ substantially in resource use (*Gotelli & Ellison, 2002*).

Competition between species is thought to be a strong force structuring ant communities (*Hölldobler & Wilson, 1990*), although recently this view has been challenged (*Cerdá, Arnan & Retana, 2013*; *Gibb & Johansson, 2011*). Competition between ant colonies involves territory defence (*Adams, 1994*; *Tanner & Adler, 2009*), nest site selection and protection (*Dejean et al., 2008*; *Fayle et al., 2015*), and competition for food resources (*Blüthgen, Stork & Fiedler, 2004*; *Blüthgen & Fiedler, 2004*; *Blüthgen, Mezger & Linsenmair, 2006*). In tropical canopy ant communities, competition sometimes results in the formation of patterns known as "ant mosaics". Ant mosaics are defined as occurring where there is a hierarchical structure, dominated by a particular set of species, which may tolerate the presence of a small number of subordinate species (generally those with less populous colonies and low levels of aggressiveness) in their territories (*Blüthgen & Stork, 2007*; *Dejean et al., 2007*). The identity of these subordinate species is sometimes determined by the identity of the dominant species present (*Room, 1971*). The dominant ant species do not tolerate each other's presence, and react aggressively to any dominant ant individual that is not from their colony (either conspecific or heterospecific). Thus, large colonies of dominant ants with defended territories can shape the pattern of ant distribution in the canopy, such that the presence of other ant colonies is not random (*Dejean & Corbara, 2003*). If these territories extend across multiple tree canopies, then the result is a "mosaic" of mutually exclusive dominant species. The formation of ant mosaics has implications for the broader ecology of forest canopies, since the identity of the dominant ant species can have impacts on other taxa, including those of functional importance (*Dejean, Djiéto-Lordon & Durand, 1997*).

Ant mosaics have been studied in many habitat types (e.g., *Blüthgen & Stork, 2007*; *Majer & Camer-Pesci, 1991*) and are often described as characteristic elements of canopy ant communities from tropical agricultural systems (*Majer, Delabie & Smith, 1994*; *Pfeiffer, Tuck & Lay, 2008*; *Room, 1971*; *Room, 1975*) and from disturbed secondary forests (*Dejean et al., 2016*; *Dejean & Corbara, 2003*). However, until recently, their presence in the canopies of undisturbed tropical forests remained contentious. Direct observations at ground level of canopy species that descend to forage in this stratum shows strong segregation between

dominant species (*Davidson et al., 2007*). Sampling the lower canopy of tropical forests using insecticide fogging does not detect any mosaic-like pattern (*Floren & Linsenmair, 2000*), while ground-based fogging targeting the canopy as a whole, the results of which are also likely influenced by lower canopy layers, also fails to detect any mosaic pattern (*Fayle, Turner & Foster, 2013*). However, canopy sampling conducted *in situ*, using a canopy crane (*Ribeiro et al., 2013*) and a range of other direct canopy access methods (*Dejean et al., 2000*; *Dejean et al., 2015*), reveals the presence of strong mosaic patterns.

Despite this progress in terms of understanding the structure of ant mosaics in the upper canopy layers in tropical rainforest, there remain outstanding questions. It is not yet known how detection of ant mosaics is affected by the sampling method used, spatial scale of sampling, or time of day at which sampling took place. For example, studies using baiting tend to find species segregation (e.g., *Ribeiro et al., 2013*), while those using fogging give more variable results, at least in primary forests, with some studies finding segregation (*Blüthgen & Stork, 2007*), and some showing random co-occurrence between species (*Fayle, Turner & Foster, 2013*; *Floren & Linsenmair, 2005*). Furthermore, ground-based fogging might not test adequately for ant mosaics in the high canopy, since this method is likely to sample a range of unconnected canopy strata, predominantly the lower layers, relating to different trees (*Blüthgen & Stork, 2007*; *Dial et al., 2006*), and leading to apparently random co-occurrence simply because many species found in a single sample would not be sufficiently connected to each other to interact. Furthermore, although ant mosaics are usually characterised by mutual exclusion of dominant species at the scale of entire individual trees, the impact of (horizontal) spatial scale of sampling on patterns of segregation is not known, in particular, whether patterns of segregation at smaller scales, within trees, differ from those between trees. Finally, although the manner in which canopy ant activity changes with time of day has been documented (e.g., *Tanaka, Yamane & Itioka, 2010*), the impact of these changes on patterns of species segregation are not known.

We tested whether using different sampling methods, restricting sampling to the uppermost canopy layer, sampling at different spatial scales (with replication both within trees and between trees), and sampling at different times of day affects patterns of species segregation. We sampled canopy ants in the same forest as *Fayle, Turner & Foster (2013)*, but rather than using ground-based fogging, we elevated both fogging machine and trays into the canopies of emergent trees (following *Ellwood & Foster, 2004*), hence ensuring that only the focal tree was sampled. Note that because individual emergent trees are usually not directly adjacent to each other, this situation differs from a number of other ant mosaic studies, where continuous areas of canopy were surveyed. The same trees were also sampled using a bait-based method during both the day and the night. We used null modelling of species co-occurrence to test for differences in patterns of species segregation at both within- and between-tree scales. Descriptions of ant community composition are presented elsewhere (*Yusah et al., 2012*; *Yusah & Foster, 2016*).

## MATERIALS AND METHODS

### Field sampling

We collected ants from 20 emergent trees (range 38.8–60.2 m height) of the genus *Parashorea* in lowland dipterocarp rain forest in Danum Valley Conservation Area (DVCA), Sabah, Malaysian Borneo (117°49′E, 5°01′N, elevation 170 m) during the periods 13 September 2007–29 February 2008 and 28 March–20 August 2008. Trees belonged to the species *P. tomentella* ($N = 5$) and *P. malaanonan* ($N = 15$), between which there were no differences in ant species composition (*Yusah & Foster, 2016*). KMY received permission to conduct fieldwork in DVCA from Danum Valley Management Committee. These trees are part of the high canopy, but emerge above the continuous canopy layer, with which they are only poorly connected. We chose to test for the presence of ant mosaics across widely spaced sampling points, rather than by sampling a smaller, continuous area. While we were unable to document the spatial extent of individual ant territories, we were nonetheless able to test for patterns of segregation between ant species that would support the existence of an ant mosaic (note that ant mosaics are defined as being mutually exclusive dominance of particular ant species across multiple adjacent canopies). Furthermore, because our sampling points were widely spaced, our results are more likely to be representative of patterns at broader spatial scales than if we had sampled a smaller, continuous area.

Two methods were used: 1. Purse-string trapping ($N = 8$ traps per tree), which allows remote collection of baited traps without disturbance of the baits prior to collection (Fig. 1A; *Yusah et al., 2012*) and 2. Canopy fogging ($N = 20$ 1 m$^2$ trays per tree) in which both trays and fogging machine were hoisted into the canopy (Fig. 1B). Both of these methods allow collection of ants from a single canopy, minimising unwanted sampling of ants from other trees, which is vital if ant communities are to be assessed for interactions. The two sampling methods are complementary, in that baiting samples the ants that dominate food resources, while fogging samples more broadly the species with foraging workers present in the canopy. Canopy work was conducted using rope access methods (Fig. 1C). Baiting was carried out both during the day and the night ($N = 8$ per time of day), near and far from the main trunk ($N = 4$ per distance) and using two different bait types: tuna and sugar water ($N = 4$ per tree). Baiting was always carried out before fogging. Analysis of community composition showed that baiting communities differed between the day and night (CCA on square root transformed abundances, permutation test: $F = 4.13$, $P = 0.001$, no. of permutations = 999), but not between bait types ($F = 1.21$, $P = 0.184$, $n$ perm = 999), or distances from the trunk ($F = 1.35$, $P = 0.090$, $n$ perm = 999). Hence, separate co-occurrence analyses were carried out for day and night samples (see below).

### Ant identification

Ants were identified to genus (*Bolton, 1994*; *Hashimoto, 2007*), and then sorted to morphospecies, with species names assigned where possible (*Bolton, 1974*; *Brown, 1978*; *Fisher, 2010*; *Pfeiffer, 2013*; *Rigato, 1994*; *Schödl, 1998*).
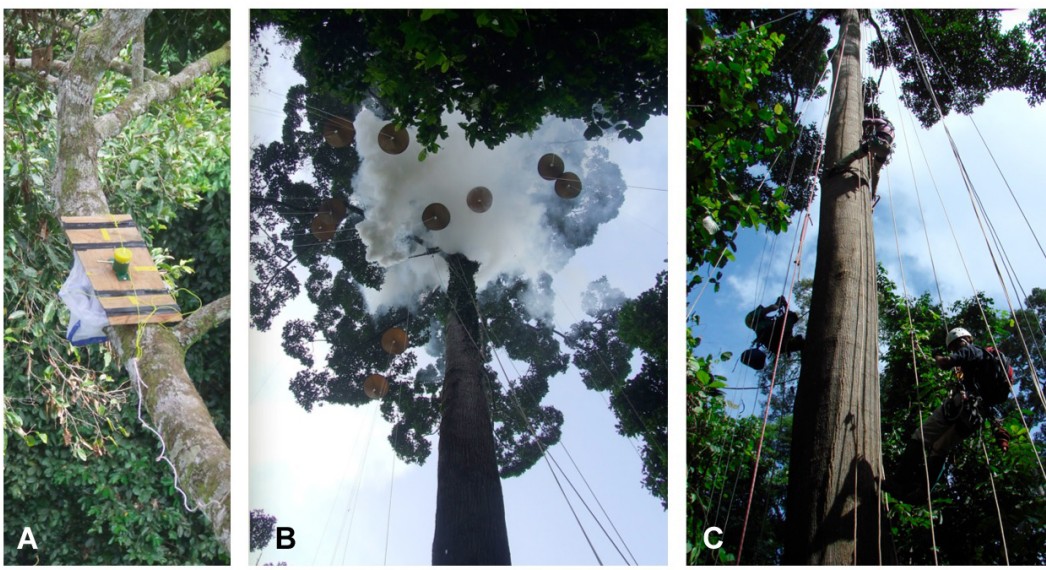

**Figure 1** Ant communities were sampled using (A) bait-based purse-string trapping and (B) fogging with fogging machine and trays hoisted into the canopy. Both methods exclusively sample ants in the crown of the focal tree and were conducted using rope access (C).

## Statistical analysis

We used null model analyses to examine whether the ant assemblages occurring in the 20 sampled trees were structured at random or in a deterministic pattern that would be consistent with the existence of an ant mosaic. Analyses were carried out at two different spatial scales to examine whether co-occurrence patterns were scale-dependent: firstly within each tree (although with results summed across all trees) and secondly between all trees, the scale that corresponds to the traditional notion of an ant mosaic. Analyses were conducted in R 2.9.2 (http://www.R-project.org; see Appendix S1 for code, Appendix S2 for data), utilising the vegan package (*Oksanen et al., 2015*). Three matrices were created comprising the ants trapped using baiting during the day and night ($N = 160$ traps each across 20 trees, two matrices), and those caught using fogging ($N = 400$ traps across 20 trees, one matrix).

We first tested for significant segregation within each tree by running within-tree analyses for all 20 trees and combining the results. This was carried out by counting the number of checkerboard units in the sub-matrix comprising the samples from each tree. Checkerboard units (CU) are counted by looking across all possible pairs of species and pairs of sampling locations (i.e., between fogging trays or purse-string traps in the case of this study). A checkerboard unit occurs when, comparing the occurrence of two species across two sampling locations, only one species occurs in each area, with each area containing a different species (*Stone & Roberts, 1990*). We summed the number of checkerboards obtained from each tree across 20 trees to give an observed within-tree chequerboard count. To obtain the distribution of checkerboard units expected if species co-occurred at random within trees, the matrix from each tree was randomised while keeping the number

of species per site and the number of occurrences per species constant. This was done using the *quasiswap* algorithm of the *commsimulator* function in R. Critically, randomisation was only conducted within each tree. All checkerboard units for the simulated data were then summed across all 20 randomised trees (in the same way as for the observed data) and this process was repeated 1,000 times. The observed number of checkerboard units was then compared to this null distribution to give the probability of obtaining that many checkerboard units or more under the null hypothesis of species co-occurring at random to each other i.e., to generate a *P*-value. Note that we did not standardise the number of chequerboard units by matrix size to give a *C*-score (*Gotelli, 2000*), because we wished to sum the total number of units across all matrices.

We then examined the pattern of assemblages across larger scales, between trees, to determine whether or not species co-occurrence between trees differed from random. Twenty "virtual trees" were made by selecting at random eight traps or 20 fogging trays (from eight or 20 trees respectively) while ensuring that there were never two or more traps from the same tree. Analysis of the 20 virtual trees was then carried out as above. The process was repeated 10 times to reduce variations caused by random sampling in the construction of virtual trees. The resulting ten *P*-values were averaged. Since we ensured that the analyses looking at within-tree and between-tree scales used exactly the same sample size and data structure, results from the first analysis and the second analysis were directly comparable, because the tests should have the same statistical power. This would not have been the case if data from all traps of one kind within a tree were combined, and analyses carried out using these much larger samples. Note that just one of these ten replicated histograms is plotted for each analysis of between-tree patterns of segregation.

## RESULTS

We found diverse ant communities in the *Parashorea* canopies (Fig. 2). For fogging (only conducted during the day), the numerically dominant ant species were *Vollenhovia* sp. 5 (most abundant species on four trees), *Dolichoderus thoracicus* (three trees), *Myrmicaria* sp. 1 (two trees), and *Crematogaster difformis* (two trees). Nine other different species numerically dominated the remaining nine trees. Baiting during the day gave a completely different result in terms of numerical dominance, with a different species *Vollenhovia* sp. 2 most abundant on seven trees. Other numerically dominant species included *Tetramorium laparum* (two trees) and *Dolichoderus magnipastor* (two trees) with the remaining nine trees being dominated by different species. *Polyrhachis ypsilon* was present on six trees, but only numerically dominant on a single tree, although simple numerical dominance probably underestimates the impact of this large-bodied species. Overall, the numerically dominant species differed between fogging and baiting for 17 of the 20 trees, with only three trees being consistent between sampling methods (inhabited by *Crematogaster difformis*, *Vollenhovia* 2, and *D. magnipastor*).

For baiting at night, *Vollenhovia* sp. 2 (the same species that was dominant during the day) was most abundant on four trees, *Dinomyrmex gigas* on three trees, and *Camponotus* sp. 2 on two trees. *Dinomyrmex gigas* was present at lower abundances (but presumably still

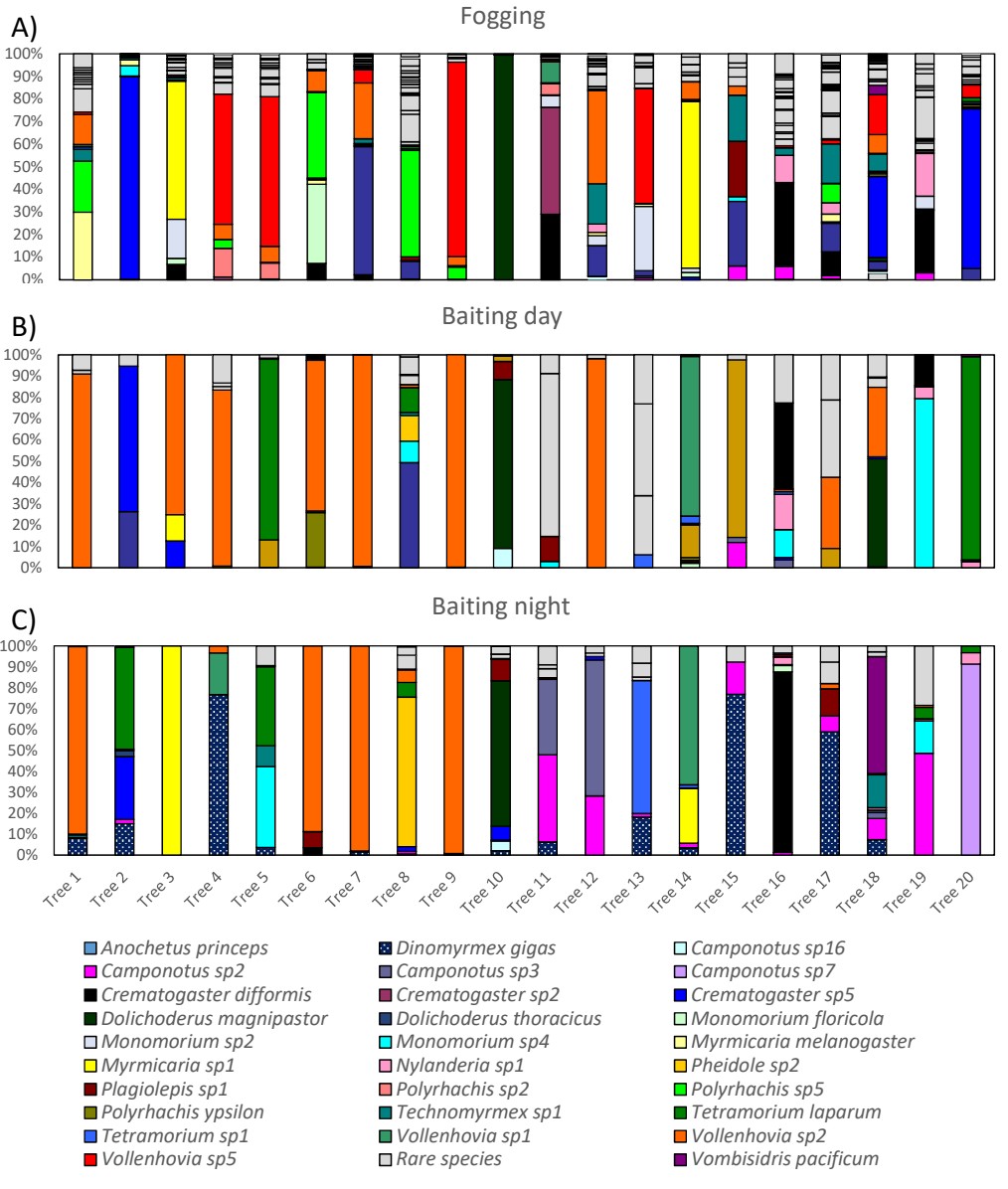

**Figure 2** **Ant communities were highly diverse, for both fogging (A) and baiting (B, C).** Different colours/patterns denote abundances of different ant species and each column shows the ant community from a single tree, with numbering of trees and colouring of ant species being consistent between different sampling methods and times of day. To avoid using many similar colours, we only coloured ant species that were either the first or second most abundant species on a tree for a particular sampling method, and were also found with an abundance of at least 50 on at least one tree. Species not meeting these criteria are coloured light grey, but are still distinguished individually by black outlines. We also coloured two large-bodied species that did not (quite) meet these criteria: *Dinomyrmex gigas* and *Polyrhachis ypsilon*.

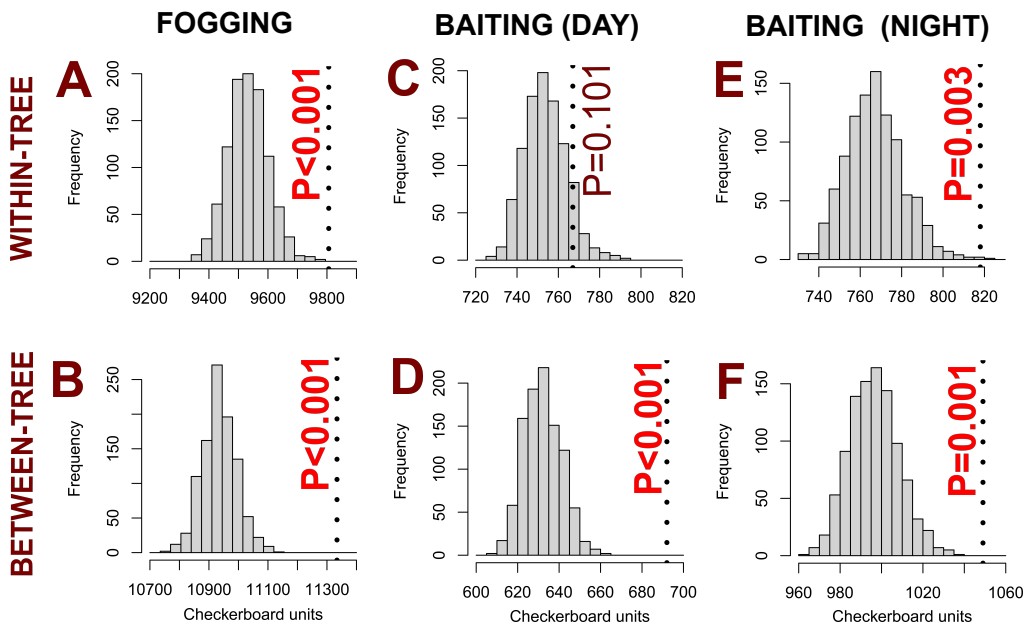

**Figure 3** **Ant communities were segregated for most combinations of scale and sampling: canopy-based fogging (A, B), baiting (day: C, D; night: E, F).** Only baited ants at within-tree scales during the day were not segregated (C). Histograms show expected distribution of chequerboard units if species co-occur at random, while vertical broken lines denote observed number of species co-occurrences. Where the vertical line is to the right of the histogram, there are more chequerboard units than expected and hence there is segregation between species (*P*-values in bold red are statistically significant). Note that for the between-tree analysis the histograms represent the results of a single example selection of sampling units to create "trees", while *p*-values have been calculated using all ten tree selections. Hence, histograms (but not *p*-values) will show minor variations if code is rerun.

high biomass, since they have large body size) on a number of other trees (14 in total). The most numerically dominant species differed between the day and the night for 12 of the 20 trees. Specifically, *Vollenhovia* sp. 2, *Dolichoderus magnipastor*, *Tetramorium laparum*, were dominant on trees during the day and also remained so at night, while in other trees, new species became more abundant at night, the most abundant of which was *Dinomyrmex gigas*. *Tetramorium laparum*, although persisting on most trees at night, was replaced by *Camponotus* sp. 7 on one tree (tree 20).

We found species segregation, consistent with the existence of ant mosaics, at between-tree scales for both fogging (Fig. 3B) and baiting (Figs. 3D and 3F). That is, there were pairs of species that co-occurred on entire trees less often than would be expected by chance. At within-tree scales, baited ants collected during the day were not segregated (Fig. 3C), that is, there were no patterns of exclusion between species for sites within the same tree. However, within trees, fogged ants (Fig. 3A) and baited ants collected during the night (Fig. 3E) were segregated. These patterns of segregation were present despite the diverse nature of the ant community, with considerable turnover of species between trees, and high species richness within individual trees during both the day and the night (for baiting) and for different sampling methods (Fig. 2).

## DISCUSSION

High-canopy mosaics at between-tree scales were detectable using both fogging and baiting. This confirms previous studies targeting sampling in the high canopy of primary forests that found strong segregation between species (*Dejean et al., 2015*; *Ribeiro et al., 2013*). Although the rarity of species precludes isolation of the particular species driving these patterns of segregation, we can speculate on possible dominants monopolising tree canopies. One candidate species for driving these patterns is *Crematogaster difformis*, a species inhabiting the myrmecophytic epiphytic ferns, *Lecanopteris* sp. and *Platycerium* sp., which has previously been demonstrated to reduce canopy ant species richness and abundance (*Tanaka, Yamane & Itioka, 2010*), and is even sufficiently dominant that it protects the fern's host tree from herbivory (*Tanaka, Inui & Itioka, 2009*). A second possible behaviourally dominant species is *Dolichoderus thoracicus*, which is known to form mosaic patterns in plantations with *Oecophylla smaragdina* (*Way & Khoo, 1991*). A second species in this genus, *Dolichoderus magnipastor*, a Bornean endemic known to tend sap-sucking Allomyrmococcini bugs (*Dill, Williams & Maschwitz, 2002*), is also likely to be a behaviourally dominant species driving these patterns (note that *Dill, Williams & Maschwitz (2002)* describe the species as diurnal, but we also collected them from baits during the night). The most abundant species for any one sampling method was *Vollenhovia* sp. 2, which was most abundant at seven trees at baits during the day. To our knowledge, *Vollenhovia* has not been found previously to be a dominant genus in the high canopy of rain forest, or any other habitat, although it has been found rarely in canopies in both plantation (*Pfeiffer, Tuck & Lay, 2008*; *Room, 1975*) and rain forest (*Floren & Linsenmair, 2000*). Finally, although present in low abundances, *Dinomyrmex gigas* has large body size, and was present on 14 of the 20 trees at night (baiting), and hence is another candidate as a dominant species in the upper canopy. Interestingly, *Oecophylla smaragdina*, a species know to be a highly dominant canopy species in primary forests elsewhere in Borneo (*Davidson et al., 2007*) was not found during sampling, and appears to be locally absent in the area surrounding Danum Valley Field Station, despite being present in nearby primary forests in Sabah (K. Yusah, T. Fayle, 2012, pers. obs.). Overall, perhaps because sampling was spread between widely spaced trees, a diverse set of species contributed to the patterns of segregation observed through a kind of "diffuse competition".

The robustness of patterns of segregation to the sampling methods employed is likely to be because dominant ants are those that are both highly abundant in general, and are also those that dominate food sources. The dominant species differed between the two sampling methods, hence demonstrating that the ants that are generally numerically dominant in the canopy (fogging samples) and those that dominate baits (baiting samples) both show patterns of segregation congruent with the existence of an ant mosaic. The difference in community composition between the sampling methods also cautions against comparison between studies using different methods; although the communities sampled using the two methods showed similar patterns of segregation, this will not necessarily always be the case. Although ants were segregated at larger scales regardless of the sampling method used, at smaller scales, within the canopies of individual trees, fogging detected patterns of
segregation, whereas baiting during the day did not. This might relate to lower statistical power of baiting, due to the smaller number of species present at each bait (mean = 1.9 species per bait for day baiting), compared with each fogging tray (mean = 5.6 species per tray). Alternatively, fogging and baiting might sample different aspects of within-tree ant spatial structuring, which seems likely, since they sample different overall communities.

*Blüthgen & Stork (2007)* suggested that rain forest canopies may be too complex for detection of mosaics after two studies (*Floren & Linsenmair, 2000*; *Floren & Linsenmair, 2005*) failed to demonstrate this pattern in the continuous canopy layer of the lower rainforest canopy using ground-based fogging of individual trees. However, our study was carried out within emergent trees where the canopies are isolated and hence are more similar to those in the plantations where ant mosaics have been observed previously (*Majer, 1976a*; *Majer, 1976b*). Therefore, the arguments by *Floren & Linsenmair (2000)* that the absence of an ant mosaic in their study was due to the habitat complexity within the canopy of tropical rainforest may still be consistent with our results. Our results also contrast with those of *Fayle, Turner & Foster (2013)* from the same forest, who did not detect species segregation using ground-based fogging. Sampling entire vertical columns of forest, as ground-based fogging does, is likely to result in collection of ants from multiple, unconnected canopy layers, and hence will not reveal patterns of segregation limited to a particular canopy stratum (*Dial et al., 2006*). Furthermore there may be a vertical gradient in strength of segregation, with segregation being strongest at the top of the canopy (*Ribeiro et al., 2013*). Finally, isolated emergent trees may be more easily defended, while the well-connected lower canopy will allow greater movement and hence potentially result in less segregation in this stratum (although note that *Dinomyrmex gigas*, which nests on the forest floor, is nonetheless able to access these tree crowns).

In terms of spatial effects, the only difference for within and between-tree scales was found for baited ants during the day, where ants were segregated between trees, but not within them. This was not an effect of differing statistical power between the analyses, since we controlled for this using our null models (see Methods). Such a pattern indicates random co-existence between a single dominant and several subordinate species within each canopy, but with different dominants occupying different canopies. The detection of segregation using the same sampling method at night might relate to the presence of *Dinomyrmex gigas* (see below). However, it is not clear why the same dependence on spatial scale is not present for fogging data, where previous work showed random co-occurrence at small and large scales for ground based fogging (*Fayle, Turner & Foster, 2013*), in contrast to the segregation at both scales observed in this study.

At within-tree scales, time of day affected patterns of segregation. Baited ants were not segregated during the day, but were at night. (Note that we do not know whether fogging at night would have resulted in the same pattern.) This difference may be driven partly by temperature. When temperatures are excessively high for ants in the canopy, the majority of ants will reduce foraging activity, including dominant ants (*Parr, 2008*), hence possibly reducing the chance of formation of mutually exclusive territories at this time. One candidate species for driving this pattern is *Dinomyrmex gigas*, an extremely large-bodied ground nesting species that maintains extensive three

dimensional territories, both at ground level, and in the rain forest canopy (*Pfeiffer & Linsenmair, 2000*; *Pfeiffer & Linsenmair, 2001*). The species tends sap-sucking hemipterans (*Pfeiffer & Linsenmair, 2007*), which are presumably clumped in their distributions, potentially leading to segregation of ant communities at small scales during the night, driven by presence of *D. gigas*. During the day *D. gigas* was not present, and within tree segregation did not occur. The presence of this large-bodied species explains how the differences in segregation between times of day might have come about despite similarities in degree of numerical dominance between times (there were 17 trees that during the day had more than 50% of the total ant abundance comprising one species, with 16 showing this pattern at night). These results indicate that patterns of segregation at small scales at food sources shift over the course of 24 h, and also caution against comparing results from different studies using these two methods at smaller scales.

## CONCLUSIONS

Taken together, our findings indicate that ant mosaics are present in the emergent trees of the high canopy of tropical rain forest in Malaysian Borneo, and that sampling targeted specifically on this forest stratum is crucial for revealing species segregation. Furthermore, spatial scale, time of day, and sampling method (the latter partly driven by vertical stratification of patterns), interact to affect detected patterns of segregation.

## ACKNOWLEDGEMENTS

We are grateful to the Royal Society's South East Asia Rainforest Research Partnership for logistic support, Yayasan Sabah for permission to conduct research, and Ed Turner for constructive comments on the manuscript. We thank Jonathan Majer and one anonymous reviewer for feedback that has greatly improved this article.

### Funding

Kalsum M. Yusah was funded by the South East Asia Rainforest Research Partnership (SEARRP), a Malaysian Ministry of Higher Education Fundamental Research Grant (FRG0373- STWN- 1/ 2014), and a Universiti Malaysia Sabah New Lecturer Grant Scheme grant (SLB0071- STWN- 2013). Tom M. Fayle was funded by a Czech Science Foundation standard grant (16-09427S). The funders had no role in study design, data collection and analysis, decision to publish, or preparation of the manuscript.

### Grant Disclosures

The following grant information was disclosed by the authors:
South East Asia Rainforest Research Partnership (SEARRP).
Malaysian Ministry of Higher Education Fundamental Research: FRG0373- STWN- 1/ 2014.
Universiti Malaysia Sabah New Lecturer Grant Scheme: SLB0071- STWN- 2013.
Czech Science Foundation: 16-09427S.

## Competing Interests

The authors declare there are no competing interests.

## Author Contributions

- Kalsum M. Yusah conceived and designed the experiments, performed the experiments, analyzed the data, wrote the paper, prepared figures and/or tables, reviewed drafts of the paper.
- William A. Foster conceived and designed the experiments, reviewed drafts of the paper.
- Glen Reynolds contributed reagents/materials/analysis tools, reviewed drafts of the paper.
- Tom M. Fayle conceived and designed the experiments, analyzed the data, contributed reagents/materials/analysis tools, wrote the paper, prepared figures and/or tables, reviewed drafts of the paper.

## Field Study Permissions

The following information was supplied relating to field study approvals (i.e., approving body and any reference numbers):

Kalsum M. Yusah received permission to conduct field work in Danum Valley Conservation Area from Danum Valley Management Committee.

## Data Availability

Data from: Ant mosaics in Bornean primary rain forest high canopy depend on spatial scale, time of day, and sampling method. DOI: 10.5061/dryad.sg1pn.

## Supplemental Information

Supplemental information for this article can be found online at http://dx.doi.org/10.7717/peerj.4231#supplemental-information.

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
