# Peer review of "Ant mosaics in Bornean primary rain forest high canopy depend on spatial scale, time of day, and sampling method"

_PeerJ, doi:10.7717/peerj.4231_

## Round 0.1 · original submission · Major Revisions

Both reviewers consider that this manuscript potentially represents a valuable contribution to the study of ant mosaics, but they also have substantial reservations about the submission in its current form. Both reviewers had issues with the general treatment of ant mosaics in the manuscript. Please address all comments made by both reviewers. (Note that the detailed comments made by Reviewer 1 about Fig. 1 actually refers to your Fig. 2.)

Reviewer 1 ·

Basic reporting

1. BASIC REPORTING
Clear, unambiguous, professional English language used throughout. OK
Intro & background to show context. No
Literature well referenced & relevant. No
Structure conforms to PeerJ standard, discipline norm, or improved for clarity.
Figures are relevant, high quality, well labelled & described. Rather No; only one figure which needs to be modified and improved.
Raw data supplied (See PeerJ policy). Yes, but one is not able to be open.

Experimental design

2. EXPERIMENTAL DESIGN
Original primary research within Scope of the journal. Yes
Research question well defined, relevant & meaningful. It is stated how research fills an identified knowledge gap. Rather No
Rigorous investigation performed to a high technical & ethical standard. Likely, but there are too many problems with Fig. 1 to assert that.
Methods described with sufficient detail & information to replicate. Yes

Validity of the findings

3. VALIDITY OF THE FINDINGS
Impact and novelty not assessed. Not the case
Negative/inconclusive results accepted. Not the case
Meaningful replication encouraged where rationale & benefit to literature is clearly stated. Not the case
Data is robust, statistically sound, & controlled. Likely Yes, but the problems with Fig. 1 do not permit us to verify this.
Conclusion well stated, linked to original research question & limited to supporting results. No
Speculation is welcome, but should be identified as such. Not the case

Comments for the author

Comments concerning the manuscript by Kalsum Yusah, William Foster, Glen Reynolds and Tom M Fayle entitled “Ant mosaics in Bornean rain forest high canopy depend on spatial scale, temporal scale, and sampling methods” (#8574).
The basic results presented in this manuscript likely bring new insights to the knowledge on the distribution of ants associated with emergent trees of an already studied Bornean rainforest and seem very interesting. The use of two techniques (fogging and bating) in the forest canopy permitted the authors to provide some information on their effectiveness; these results are within the scope of the journal PeerJ.
Nevertheless, several important modifications are needed.
The definition of “ant mosaics” is misinterpreted, likely under the influence of several authors working in the Neotropics (but see Ribeiro et al. 2013 cited in the text), and by the last author in Borneo. Furthermore, the bibliography is outdated and very incomplete, so that I provide the authors with a short explanation and several useful references (with the links to permit a new version to be written as quickly as possible). Note that the notion of an ant mosaic has been described in four syntheses (Leston 1973; Majer 1993, Blüthgen & Stork 2007; Dejean et al. 2007), with many explanatory facts and many useful references in the two last papers.

Explanation of what is an “ant mosaic”
The notion of arboreal “ant mosaic” was formulated by Room (1971) (cited in the text) who worked on African cocoa plantations. Meanwhile, several British researchers did the same in different African tree crop plantations (among the most recent, see Jackson 1984). This phenomenon is obvious to observe for two main reasons. (1) All of these plantations were composed of small trees, so that the ant mosaic was “in front of the observers’ eyes”. (2) The “territorially dominant arboreal ant species” (TDA ants), with their huge colonies, are easy to determine as the three main genera, Oecophylla, Tetramorium and Crematogaster, have workers with very typical morphologies. Also, Oecophylla is monospecific, Tetramorium is mostly represented by T. aculeatum, and the workers of the different Crematogaster species are not very similar (at least at the local scale). Furthermore, the nests of these ant species are very different from each other, rendering the identification of the different Crematogaster species easy. Voucher specimens of these ant species were deposited in the Museum of Natural History, and Brian Taylor and Barry Bolton (both systematicians) helped ecolo-agronomists when necessary.
Among these researchers, Bigger (1993) showed that conspecific trees of different ages sheltered different dominant ant species (introducing the idea of ontogeny; see Dejean et al. 2016 and papers cited therein). Also, the biology, behavior and ecology of many of these TDA ant species have been studied by different authors, more particularly their territoriality (markings that persist several years: Hölldobler & Wilson 1978; Beugnon & Dejean 1992; Offenberg 2007) and nest site selection (Djiéto & Dejean 1999). Ant mosaics exist in plantations (i.e., presented by Leston in 1973; see also Majer et al. 1994), forest edges, and secondary and old forests (see Blüthgen & Stork 2007; Davidson et al. 2007; Dejean et al. 2007, 2015; Ribeiro et al. 2013).
Note that arboreal ant mosaics exist in Bornean forests (see Davidson et al. 2007) with the fantastic nocturnal, giant ant species Camponotus gigas whose nests are at ground level so that the workers need to climb up the 60-m tall trunks of dipterocarps to forage in their crowns; also, they are territorial (see Pfeiffer & Linsermair 1998, 2001).
Because the authors worked on several ant species able to share a territory with a TDA ant species, they need to state precisely what species are numerically dominant with the aim of detecting territorially dominant species (see definitions in Davidson 1998). “Co-dominant” species can be determined through comparisons, and “non-dominant” species are generally the remaining species with comparatively small colonies. Note that co-dominants correspond frequently to a nocturnal and a diurnal species sharing one or several adjacent tree crown(s) over the nycthemeron (details in Dejean et al. 2007); for example, this is the case for trees N°4, 15 and 17 in “baiting” (Fig.1G and 1H) where the nocturnal species Camponotus gigas is involved.

Comments directly concerning the manuscript
Introduction
This section needs to be entirely rewritten.
- Lines 42-50. It would be better to present the true objectives of the study (i.e., related to two kinds of comparisons: during the daytime, fogging versus baiting; baiting diurnally versus at night) before presenting the seminal studies by Nicholas Gotelli. The relationship between these objectives and the notion of ant mosaics can be presented after along with the definition of arboreal ant mosaics. More importantly, the authors need to define what are territorially dominant ant species and co-dominant species (the latter, for example, because of the presence of the nocturnal Camponotus gigas).
- Lines 51-55. The notion of competition likely misled the authors because ant mosaics are the result of the territoriality of the TDA ant species. The reactions which trigger territoriality here are related to the colony closure, and so, to colony mate recognition. Discrimining colony mates from aliens is based on the comparison of the cuticular hydrocarbons of members of the colony (i.e., these hydrocarbons make up the colony odor which serves as a “template”) with those of encountered ants, a mismatch usually results in aggressiveness (see the synthesis by Bos & d’Ettore 2012).
- Line 59, Dejean & Corbara (2003) is outdated; rather cite Blüthgen & Stork (2007) and/or Dejean et al. (2007).
- Line 61. “Thus it is “possible” ….. This part needs to be entirely rewritten.
- Lines 69-71. “However, their presence in undisturbed tropical forests remains contentious, with ant species being found to be segregated in some studies (e.g. Ribeiro et al. 2013) but not others (Fayle et al. 2013; Floren & Linsenmair 2000)”. This sentence presents flagrant untruths. (1) Several papers other than Ribeiro et al. (2013) deal with the presence of an ant mosaic in old forests (see Dejean et al. 2007 and papers cited therein; Blüthgen & Stork 2007, which is cited in the present manuscript; Davidson et al. 2007 presenting a study conducted in a Bornean forest; Dejean et al. 2015, 2016 for studies conducted in Africa). (2) The paper by Floren & Linsenmair (2000) needs to be read, not only the title, because these authors worked in the lower canopy where they did not find a mosaic as did Ribeiro et al. (2013), so that these two papers are congruent (i.e., they cannot be placed in opposition); Blüthgen & Stork (2007) also showed that the presence of an ant mosaic depends on the stratum studied (absent from the lower canopy). Concerning the paper by Fayle et al. (2013), which is very interesting overall, the approach using separate trees cannot really permit the detection of an ant mosaic (even if the model permits one to see an effect in plantations through the relationships between ants and an epiphytic fern).
- Lines 72-75. Contrarily to what is asserted, fogging can permit to show the presence of a mosaic, but on the condition that the authors work on numerous, adjacent trees (see Watt et al. 2003; Blüthgen & Stork 2007). Working on a small number of distant trees is another approach which is particularly useful for studying ant species diversity, the relationships between a tree species and arboreal ants etc.
Note that when fogging is conducted along a transect, when it reaches the territory of TD ant species “A”, a rain shower of numbed “A” workers falls to on the ground under several trees until the fogging reaches the territory of species “B”, etc. So, the objectives of working on isolated trees are very different from those of working along a transect.

Materials and Methods
- Line 95. Everyone working in tropical forests can easily obtain the identification of at least the emergent species; this is likely the case for the genus Parashorea in a Bornean dipterocarp rainforest. The species name might be useful for certain readers (i.e., congeneric trees of different species can attract ant queens and workers from different species, the resulting nest-site selection can therefore influence the ant mosaic).
- Lines 112-115. It is a shame that at least the most frequent dominant ant species were not identified to species level. This was likely possible for Tetramorium sp.2 (using the collections of the Museum of Natural History, London); the same is true for Myrmicaria spp. and maybe for certain Vollenhovia species (likely deposited in German collections and/or the Museums of Los Angeles, San Francisco and Harvard).
- Lines 122-123. The first Supplemental Appendix is not lisible.

Results
- Fig. 1 needs to be clarified: (1) several colors representing ant species presented in the reference grids appear as a nuance of grey in the figure itself; (2) the colors representing the same ant species through baiting and fogging need to be the same. In other words, to present only one reference grid with the numerically dominant species easily distinguishable from each other at a glance is strongly needed (this is less important for the other species). Some of the colors in Fig. 1 are not represented in the reference grid (ex. the yellow from trees 5 and 15 in Fig. 1G differs from that concerning Polyrhachis ypsilon, but is rather similar to that of Dolichoderus sp. 7 from the reference grid in Fig. 1I)
- I therefore suggest that the authors thoroughly verify all correspondences between the colors selected and the ant species. I indeed suspect some mistakes in the Fig. 1 presented here.
- Readers need to be absolutely sure that the trees examined through baiting (Fig 1D & H) are exactly the same as those examined through fogging (Fig. 1I), and that the sequences of 20 trees in both cases are exactly the same (i.e., tree N° 1 in baiting is also tree N°1 in fogging etc.). This is likely the case but see the last comment above (I suspect some mistakes). If not, this needs to be stated in the Materials and Methods.
- If the sequence of trees is the same, it would be better to present “fogging” first (so, Fig. 1I) and then “baiting” (so, Fig. 1G and then Fig. 1H) and, of course, to change the codes for the figures. Consequently, baiting diurnally will become the reference for both fogging (also occurred diurnally) and baiting nocturnally, something facilitating comparisons in the new Fig. 1.
- Comparison between fogging and baiting (diurnally). It is normal that fogging resulted in many more ant species per tree than baiting. Fogging kills workers of all ant species occurring in the zone of the tree crown treated, while baiting selects (1) species attracted by the baits, and (2) among them, those that arrive first or those that have excluded the other species. Therefore, comparing fogging and baiting on the same tree crowns (diurnally) is interesting as it can permit to determine, among numerically dominant species, those which have the skill to eliminate competitors from the baits (i.e., likely the most territorial). Yet the fact that many colors are replaced by nuances of grey plus others that do not exist in the reference grid renders this comparison difficult in the current Fig. 1.
- Comparison between baiting diurnally and at night. Baiting diurnally (Figs. 1A, 1G) illustrates the presence of numerically dominant species on each tree (we cannot say that they are “territorial” without a specific study on each species). That the model does not show a significant difference for individual trees is not a problem (if we take the reasoning further, only one TDA ant species can occupy an entire tree crown which is in accordance with the notion of an ant mosaic; the case for tree N°9). The comparison with baiting diurnally and baiting at night (Figs. 1B, 1H) is interesting. For example, tree N°4 was occupied by two co-dominant species: Vollenhovia sp.2 was dominant both day and night in trees N°6, 7, and 9, while it was present diurnally in tree N°4 and pushed away from the baits at night by the nocturnal Camponotus gigas. In tree N°19 Tetramorium sp.2 is replaced at night by Camponotus sp.19, but here it is possible that the former is diurnal, the latter nocturnal (so the interest of identifying the ants to species level because the biology of many of them has been studied by German and American researchers).

Discussion
- Lines 166-192. The Discussion, is based from the start on the idea that mosaics do not exist in tropical rainforest canopies, but this is not the case, so that this part needs to be deleted or, maybe, rewritten based on the information I provided above.
- Lines 193-210. This study rather deals with (1) the comparison of two techniques (during the daytime) and (2) one technique (during the daytime and at night). The Discussion needs to concentrate first on the interests of these comparisons in greater depth than currently presented.
The indirect advantage of the results acquired through these techniques can then be presented in light of the notion of ant mosaics. This implies that the authors thoroughly verify that the colors in Fig. 1 correspond to the different ant species they recorded. The comparison between diurnal fogging and baiting will likely provide information on what species dominate the others at baits among those the most represented during fogging (i.e., numerically dominant species).

In conclusion
I followed to the letter the instructions concerning reviewing a manuscript for the journal PeerJ, so that I tried to provide the authors with information permitting them to submit a new version of their manuscript that is in accordance with their true aim and with a correspondence between the latter and the notion of ant mosaics.
The core results of this study are likely much more interesting than the authors realise. They need to rethink Fig. 1 and verify that the different ant species represented are really those observed during the study (i.e., a thorough check is necessary).
The Introduction and the Discussion need to be entirely rewritten. With only 20 trees isolated from each other, an ant mosaic is possible to be suspected, but this needs to be explained based on the results (see above). Indeed, the authors rather showed the presence of numerically dominant species, the existence of co-dominant species sharing the same territory (one during the day the other at night), that one species can exclude another diurnal species at baits during the night, etc. The comparison between fogging and baiting (during the daytime) could permit them to show the domination at the baits by certain species (but Fig. 1 with the same colors both in the PDF and in the Microsoft Office Picture Manager needs to be improved to permit readers to really understand).

References cited (with online links except for Leston 1973; I added Blüthgen & Stork 2007 as I am not sure that all of the authors have read it).
Beugnon G, Dejean A 1992. Adaptive properties of the chemical trail system of the African weaver ant. Insectes Sociaux 39: 341–346. http://download.springer.com/static/pdf/361/art%253A10.1007%252FBF01323954.pdf?originUrl=http%3A%2F%2Flink.springer.com%2Farticle%2F10.1007%2FBF01323954&token2=exp=1454352742~acl=%2Fstatic%2Fpdf%2F361%2Fart%25253A10.1007%25252FBF01323954.pdf%3ForiginUrl%3Dhttp%253A%252F%252Flink.springer.com%252Farticle%252F10.1007%252FBF01323954*~hmac=79b5100cc58b4b725e4be5613d848f357b60c5442e6184ac279089e1f9baeb35
Bigger M 1993. Ant-hemipteran interactions in a tropical ecosystem. Description of an experiment on cocoa in Ghana. Bulletin of Entomological Research 83: 475–505. https://www.researchgate.net/publication/231887655_Ant_-_homopteran_interactions_in_a_tropical_ecosystem_Description_of_an_experiment_on_cocoa_in_Ghana
Blüthgen N, Stork NE 2007. Ant mosaics in a tropical rainforest in Australia and elsewhere: a critical review. Aust Ecol 32:93–104. http://onlinelibrary.wiley.com/doi/10.1111/j.1442-9993.2007.01744.x/pdf
Bos N & d’Ettorre P 2012. Recognition of social identity in ants. Frontiers in Psychology 3: 83. http://www.ncbi.nlm.nih.gov/pmc/articles/PMC3309994/pdf/fpsyg-03-00083.pdf
Brühl CA, Gunsalam G, Linsenmair KE 1998. Stratification of ants (Hymenoptera, Formicidae) in a primary rain forest in Sabah, Borneo. Journal of Tropical Ecology 14: 285–297. http://journals.cambridge.org/download.php?file=%2F9022_7BA66FE54764E52B55F8A0213E95C87B_journals__TRO_TRO14_03_S0266467498000224a.pdf&cover=Y&code=87b4de4d79ce57bd329e7a33e0a9df0e
Davidson DW 1998. Resource discovery versus resource domination in ants: A functional mechanism for breaking the trade off. Ecological Entomology 23: 484–490. http://onlinelibrary.wiley.com/doi/10.1046/j.1365-2311.1998.00145.x/pdf
Davidson DW, Lessard J-P, Bernau CR, Cook SC 2007. Tropical ant mosaic in a primary Bornean Rain Forest. Biotropica 39: 468–475. http://onlinelibrary.wiley.com/doi/10.1111/j.1744-7429.2007.00304.x/pdf
Dejean A, Azémar F, Céréghino R, Leponce M, Compin A, Corbara B, Orivel J & Compin A 2016. The dynamics of ant mosaics in tropical rainforests characterized using the Self-Organizing Map algorithm. Insect Science DOI: 10.1111/1744-7917.12208. http://www.academia.edu/12887808/The_dynamics_of_ant_mosaics_in_tropical_rainforests_characterized_using_the_Self-Organizing_Map_algorithm
Dejean A., Ryder S., Bolton B., Compin A., Leponce M., Azémar F., Céréghino R., Orivel, J., Corbara B. 2015. How territoriality and host-tree taxa determine the structure of ant mosaics. The Science of Nature 102: 33. http://download.springer.com/static/pdf/936/art%253A10.1007%252Fs00114-015-1282-7.pdf?originUrl=http%3A%2F%2Flink.springer.com%2Farticle%2F10.1007%2Fs00114-015-1282-7&token2=exp=1454351993~acl=%2Fstatic%2Fpdf%2F936%2Fart%25253A10.1007%25252Fs00114-015-1282-7.pdf%3ForiginUrl%3Dhttp%253A%252F%252Flink.springer.com%252Farticle%252F10.1007%252Fs00114-015-1282-7*~hmac=1b848028f339748bb0cc4c30c501e9922c6025ae04a057e2a9c0f4cbbe5a7c9c
Dejean A, Corbara B, Orivel J, Leponce M 2007. Rainforest canopy ants: the implications of territoriality and predatory behavior. Functional Ecosystems and Communities 1: 105–120. http://www.globalsciencebooks.info/Online/GSBOnline/images/0712/FEC_1%282%29/FEC_1%282%29105-120o.pdf
Djiéto-Lordon C, Dejean A 1999a. Tropical arboreal ant mosaic: innate attraction and imprinting determine nesting site selection in dominant ants. Behavioral Ecology and Sociobiology 45: 219–225. http://download.springer.com/static/pdf/131/art%253A10.1007%252Fs002650050556.pdf?originUrl=http%3A%2F%2Flink.springer.com%2Farticle%2F10.1007%2Fs002650050556&token2=exp=1454353055~acl=%2Fstatic%2Fpdf%2F131%2Fart%25253A10.1007%25252Fs002650050556.pdf%3ForiginUrl%3Dhttp%253A%252F%252Flink.springer.com%252Farticle%252F10.1007%252Fs002650050556*~hmac=443f826d6f1da9be8eaa518d0d8b0db1076a770d4712ce7f5f30573fe9d22e02
Djiéto-Lordon C, Dejean A 1999b. Innate attraction supplants experience during host plant selection in an obligate plant-ant. Behavioural Processes 46: 181–187. http://ac.els-cdn.com/S0376635799000327/1-s2.0-S0376635799000327-main.pdf?_tid=085e5d8c-c913-11e5-a2bc-00000aab0f6c&acdnat=1454352108_ccd4ac0c8187a047f20dff11106e7d95
Hölldobler B, Wilson EO 1977. Colony-specific territorial pheromone in the African weaver ant Oecophylla longinoda (Latreille). PNAS 74: 2072–2075. http://www.pnas.org/content/74/5/2072.full.pdf
Jackson DA 1984. Ant distribution patterns in a Cameroonian cocoa plantation: investigation of the ant-mosaic hypothesis. Oecologia 62: 318–324. http://download.springer.com/static/pdf/822/art%253A10.1007%252FBF00384263.pdf?originUrl=http%3A%2F%2Flink.springer.com%2Farticle%2F10.1007%2FBF00384263&token2=exp=1454353271~acl=%2Fstatic%2Fpdf%2F822%2Fart%25253A10.1007%25252FBF00384263.pdf%3ForiginUrl%3Dhttp%253A%252F%252Flink.springer.com%252Farticle%252F10.1007%252FBF00384263*~hmac=74207aba82b785cfe88d08264e8b3b676fd4f784ac51ac67adf0acf00d66eb68
Leston D 1973. The ant mosaic, tropical tree crops and the limiting of pests and diseases. Pest Articles and News Summaries 19: 311–341. https://www.researchgate.net/publication/248979758_The_Ant_Mosaic_-_Tropical_Tree_Crops_and_the_Limiting_of_Pests_and_Diseases
Majer JD 1993. Comparison of the arboreal ant mosaic in Ghana, Brasil, Papua New Guinea and Australia: its structure and influence of ant diversity. In: LaSalle J, Gauld ID (eds) Hymenoptera and Biodiversity, CAB International, Wallingford, UK, pp 115–141.
Majer J, Delabie JHC, Smith RB 1994. Arboreal ant community patterns in Brazilian cocoa farms. Biotropica 26: 73–83. http://www.sidalc.net/repdoc/a3636i/a3636i.pdf
Offenberg J 2007. The distribution of weaver ant pheromones on host trees. Insectes Sociaux 54: 248–250. http://www.biologie.uni-ulm.de/antnet/pdf/offenberg-2007.pdf
Pfeiffer M, Linsenmair KE 1998. Polydomy and the organization of foraging in a colony of the Malaysian giant ant Camponotus gigas (Hym./Form.). Oecologia 117: 579–590. http://www.antbase.net/pdf/pfeiffer-1998.pdf
Pfeiffer M, Linsenmair KE 2001. Territoriality in the Malaysian giant ant Camponotus gigas (Hymenoptera/Formicidae). Journal of Ethology 19: 75-85. http://download.springer.com/static/pdf/410/art%253A10.1007%252Fs101640170002.pdf?originUrl=http%3A%2F%2Flink.springer.com%2Farticle%2F10.1007%2Fs101640170002&token2=exp=1454363869~acl=%2Fstatic%2Fpdf%2F410%2Fart%25253A10.1007%25252Fs101640170002.pdf%3ForiginUrl%3Dhttp%253A%252F%252Flink.springer.com%252Farticle%252F10.1007%252Fs101640170002*~hmac=6273a28a4fed4db5229746d56d743d42672bfb2344d146b9e12abf891c086938
Room PM 1971. The relative distribution of ant species in Ghana's cocoa farms. Journal of Animal Ecology 40: 735–751. http://www.jstor.org/stable/3447?seq=1#page_scan_tab_contents
Schulz A, Wagner T 2002. Influence of forest type and tree species on canopy ants (Hymenoptera: Formicidae) in the Budongo Forest Reserve, Uganda. Oecologia 133: 224–232. http://download.springer.com/static/pdf/212/art%253A10.1007%252Fs00442-002-1010-9.pdf?originUrl=http%3A%2F%2Flink.springer.com%2Farticle%2F10.1007%2Fs00442-002-1010-9&token2=exp=1454353668~acl=%2Fstatic%2Fpdf%2F212%2Fart%25253A10.1007%25252Fs00442-002-1010-9.pdf%3ForiginUrl%3Dhttp%253A%252F%252Flink.springer.com%252Farticle%252F10.1007%252Fs00442-002-1010-9*~hmac=c8ddb54646c191369135f2047c54ad22194b2bd31209b71bc1cf3bafa204e880
Watt AD, Stork NE, Bolton B 2002. The diversity and abundance of ants in relation to forest disturbance and plantation establishment in southern Cameroon. Journal of Applied Ecology 39: 985–998. http://onlinelibrary.wiley.com/doi/10.1046/j.1365-2664.2002.00699.x/pdf

·

Basic reporting

Line 58 I would specify ‘tropical’ canopies as arboreal ants are not generally abundant enough in temperate canopies for mosaics to form.

The section at lines 72-74 needs to be recast along the following lines:
• Mosaics have frequently been demonstrated in plantations;
• Floren & Linsenmair 2000 suggested that they do not exist in forest canopies, but their sampling was in lower canopy;
• Rebeiro et al. 2013 sampled the canopy in Panama from above and found strong evidence of a mosaic;
• Rebeiro’s data was tentative as it was from a short-term study;
• However, Dejean et al 2015 (an important paper that has not been cited), working in Gabon and also sampling the canopy from above, provided a more substantial data set that demonstrated the existence of a mosaic.

Alain Dejean, Suzanne Ryder, Barry Bolton, Arthur Compin,
Maurice Leponce, Frédéric Azémar, Régis Céréghino, Jérôme Orivel,
Bruno Corbara (2015) How territoriality and host-tree taxa determine the structure
of ant mosaics. Sci Nat (2015) 102:3 DOI 10.1007/s00114-015-1282-7

I believe that it is essential to take this approach to the lead in, as the story about mosaics in upper canopy has already started to unfold, and this manuscript is building upon these findings, not discovering them.

Line 174. I disagree with this statement. Surely, the fact is that your findings confirm that Rebeiro et al. and Dejean et al. are correct in suggesting that there is a mosaic in the upper canopy and that Floren and Linsenmair and Fayle et al. were probably incorrect in suggesting that it does not exist there. Floren and Linsemair’s sampling was NOT a representation of what is found in the upper canopy as they approached the canopy from beneath and did not reach the highly insolated upper layers of the canopy.

Figure 2 I don’t know if this is a problem with my computer alone, but many of the bars on the histograms are in various shades of grey, not the colours indicated by the keys.

Experimental design

Line 100 This paragraph should mention the sampling dates as there is a degree of seasonal variation in canopy ant activity, even in the tropics. Also, mention that fogging was (presumably) performed after the trees were baited.

Validity of the findings

Line 212. The lack of segregation within trees during the day is an intriguing matter. However, the analysis does not distinguish between dominant and non-dominant species. The mention of Camponotus gigas is of note, but were correspondingly abundant ants from other species not abundant in the day? Might it not be worth mentioning somewhere which of the sampled species are likely to be dominant species?

Comments for the author

Although the findings in this ms are valid, the emphasis needs to be changed in part of the Introduction and again in the Discussion. The finding of a mosaic in the upper canopy of rainforest is no longer novel, although this ms is refining what we know about mosaics in the is stratum of the forest.

I believe that the emphasis of the ms should be changed to reflect the fact that we now know that mosaics do exist in the upper canopy and that this ms sets out to provide further information on the interactions that exist in such areas.

---

## Round 0.2 · Minor Revisions

Hi, Reviewers 1 and 2 gave both made suggestions/requests for minor revisions. Reviewer 1's comments are extensive, but has several points that need addressing.

1. delete citation of Sanders 2007 in the Introduction (see comment at end of 'expt design')
2. change of name to Dinomyrmex gigas, see revision/citation in zootaxa
3. Change of colour in fig2 so that Dinomyrmex gigas can be separated from others similar colours; and perhaps some specific comment on this species
4. size of tree, in methods
5. several points in the discussion that could be looked at and you may wish to change ms to improve

Reviewer 1 ·

Basic reporting

1. Basic reporting

The manuscript by Kalsum M. Yusah et al. entitled “Ant mosaics in Bornean rain forest high canopy depend on spatial scale, time of the day, and sampling” is original because it deals with emergent trees of the Bornean rainforest and their relationships with ants. This approach, to the best of my knowledge, has not previously been used by researchers working on ant distribution in tropical rainforests.
The authors used a double, complementary methodology to conduct this study. First, only during the daytime, fogging was limited to the level of the tree crown (to catch ants from this area and not an entire vertical column as in many other studies for which this technique was used). This approach thus permits to have an ‘image’ of what ant species are active when the insecticide is itself active. Second, both during the daytime and at night, baiting was used permitting the authors to gather ant species during these two periods of the nychthemeron, and thus, ant species that were present on the baits when the baits were collected. Indeed, depending on their rhythm of activity and hierarchical position, different ant species succeed one another on the baits, certain ant species being fast at discovering a bait, but others displace them to occupy these baits (i.e., opportunists versus extirpators).
Therefore, the authors obtained precise information on the activity of ants from emergent tree crowns in the very interesting context of the Bornean rainforest.
Because the approach is very, very interesting and original, the experiment apparently well conducted as are the statistics, the rest of my comments will be aimed at improving the presentation of the core results keeping in mind that ants are social insects as I did in my comments concerning the former manuscript. Furthermore, the base study is likely much more interesting than the authors may have expected because in Bornean rainforests one of the dominant ant species is Dinomyrmex gigas whose colonies are ground nesting and polydomous, and the workers, among the biggest in the world, tend hemiperans only at night. Yet, I cannot be very helpful on this subject as the colors in Fig. 2 did not permit me to distinguish this species from four others.
Indeed, although it is not obligatory to present that in detail, the approach chosen by the authors needs to take into consideration nest site selection, polyethism, rhythm of activity, nestmate recruitment and interspecific competition particularly at baits. For the latter case I suggest the authors read papers on ground-foraging ants, including invasive species as this will permit them to “see” that several ant species succeed one another at baits.
Another suggestion: modify the title. I am not sure that the authors really adhere to the concept of ‘ant mosaics’ that deals with the territoriality of ‘territorially dominant arboreal ant species’, and so the distribution of their territories in the forest canopy. Here, the key words are rather ‘emergent tree crowns’, ‘numerically dominant species’, ‘rhythm of activity’, and, indirectly, ‘interspecific competition’.

Thus, this new version of the manuscript is much more elaborated than the previous. Only some modifications are needed to improve it and, in my opinion, the core results can permit the authors to present a much more interesting manuscript than they may think. It is a shame that Fig. 2 does not allow readers to distinguish Dinomyrmex gigas from four other ants species, so that I cannot help the authors in the presentation of the impact of this ant species.

Experimental design

2. Experimental design

The experimental design is fine on the condition that the authors admit that they did not really work on ‘ant mosaics’. I suggest that they read papers by British researchers working on cocoa plantation with shade trees, a situation that presents some similarities with emergent trees above the forest canopy (see details below). They will see that the ‘ant mosaic’ in the cocoa canopy is influenced by the presence of the ‘tall’ shade trees because certain ant species (of the genus Crematogaster) rather nest in their crowns and occupy the crowns of the cocoa trees surrounding them.
Also, the statistical approach is fine (I asked a specialist to verify that in detail for the first review). Indeed, in a few papers dealing with ant mosaics, the authors mistakenly used a wrong approach, so that they found an absence of ant mosaic where it was obviously present. This is presented in Blüthgen and Stork (2007) and Dejean et al. (2007). I therefore suggest that the authors delete the citation for Sanders et al. (2007) who used this wrong approach (see lines 62, 67, 386) in order to avoid mistakes by readers. Yet, the authors need to remember that they did not work directly on an ant mosaic (and their core results are interesting).

Validity of the findings

3. Validity of the findings

The core findings are very interesting as explained in the general comments. Only ‘cosmetic’ modifications to the text are needed, although relatively numerous.

Comments for the author

General comments for the author

In this part, I shall try to help the authors to write a definitive version of their manuscript.
Before all, I suggest that the authors re-read the main comment I made in the previous review: “The definition of “ant mosaics” is misinterpreted, likely under the influence of several authors working in the Neotropics (but see Ribeiro et al. 2013 cited in the text), and by the last author in Borneo. Furthermore, the bibliography is outdated and very incomplete, so that I provide the authors with a short explanation and several useful references (with the links to permit a new version to be written as quickly as possible). Note that the notion of an ant mosaic has been described in four syntheses (Leston 1973; Majer 1993, Blüthgen & Stork 2007; Dejean et al. 2007), with many explanatory facts and many useful references in the two last papers.”
Although this new version of the manuscript needs only cosmetic changes, the authors will understand why I suggest rewriting the title and modifying several parts of the text (see Abstract / Discussion: lines 38-40).

Abstract
The abstract and different parts in the text: avoid the word “recently” when you cite papers more than 1-year old (and particularly when they are 10 to17 years-old, see below). See lines 23, 54, 73.

Lines 22-23. It is likely that the paper by Dejean et al. 2000 was the first that was conducted on a pristine forest; Nico Blüthgen published a paper shortly after (Blüthgen et al. 2004), the Diane Davidson (Davidson et al. 2007; both are cited in the text). So, reconsider this sentence.
Dejean A., McKey D., Gibernau M. & Belin-Depoux M. 2000. The arboreal ant mosaic in a Cameroonian rainforest. Sociobiology 35: 403-423.

Introduction

Lines 82-88. Better to delete this part. Indeed, ant mosaics deal with territories, so the latter need to be delimited. The other approaches are interesting, but in certain cases do not deal really with ant mosaics (see the part in green above). I suggest that the paragraph lines 82-97 be slightly modified to present the different approaches permitting readers to understand how ants are distributed in the rainforest canopies (no more).
All throughout the text, enter Dinomyrmex gigas rather than Camponotus gigas (see Ward et al. 2016).
Ward P.S., Blaimer B.B., Fisher B.L. 2016. A revised phylogenetic classification of the ant subfamily Formicinae (Hymenoptera: Formicidae), with resurrection of the genera Colobopsis and Dinomyrmex. Zootaxa 4072: 343-357.

I suggest that the authors first revise their Fig. 2 to permit readers to distinguish Dinomyrmex gigas from Crematogaster difformis, Dolichoderus magnipastor, Plagiolepis sp.1 and Crematogaster sp.5.
If Dinomyrmex gigas appears only in a very few situations (likely during the nighttime), I suggest that the authors write a small paragraph on this ant species (see papers by Martin Pfeiffer below). If Dinomyrmex gigas is well represented, I suggest that the authors write a specific paragraph on the originality of the situation as workers of this species climb up these very tall trees each night to tend hemipterans. Thus, these hemipterans are tended by other ant species during the daytime. Therefore, Dinomyrmex gigas plays a major role in the nocturnal hierarchy of ant species (an allusion to the notion of ant mosaic can be made, but cautiously).
Pfeiffer M., Linsenmair K.E. 1998. Polydomy and the organization of foraging in a colony of the Malaysian giant ant Camponotus gigas (Hym. - Form.). Oecologia 117: 579-590. (directly available online: https://www.researchgate.net/publication/235684413_Polydomy_and_organization_of_foraging_in_Camponotus_gigas)
Pfeiffer M., Linsenmair K.E. 2000. Contributions to the life history of the Malaysian giant ant Camponotus gigas (Hymenoptera - Formicidae). Insectes Sociaux 47: 123-132. (directly available online: http://www.biologie.uni-ulm.de/antnet/pdf/pfeiffer-2000.pdf)
Pfeiffer M., Linsenmair K.E. 2001. Territoriality in the Malaysian giant ant Camponotus gigas (Hymenoptera/Formicidae). Journal of Ethology 19: 75-85. (directly available online: http://systax-vm4.biologie.uni-ulm.de/pdf/pfeiffer-2001.pdf)
Pfeiffer M., Linsenmair K.E. 2007. Trophobiosis in a tropical rainforest on Borneo: Giant ants Camponotus gigas (Hymenoptera: Formicidae) herd wax cicadas Bythopsyrna circulata (Auchenorrhyncha: Flatidae). Asian Myrmecology, 1, 105–119. (directly available online: http://www.asian-myrmecology.org/publications/pfeiffer-linsenmair-am-01-10.pdf)

Materials and Methods
First paragraph: please provide the size of the canopy trees and that of the emergent trees on which you worked to permit readers to visualize the situation (and consider at its true value the work undertaken to conduct this study).

Lines 117-118. ’(although given the lack of connectivity, it seems unlikely that these territories often extended to other trees)‘ I suggest deleting this part as it is a supposition (i.e., speculative). If we refer to shade trees in cocoa plantations, so a situation presenting some similarities with emergent trees and the rest of the rainforest canopy, one can see that some ant species nest specifically in shade trees, influencing the mosaic in the cocoa plantations around these trees (Majer 1976; there are other studies showing this situation such as those by Brian Taylor).
Majer J.D. 1976. The ant mosaic in Ghana cocoa farms: further structural considerations. Journal of Applied Ecology 13: 145–156. (directly available online: https://www.jstor.org/stable/pdf/2401934.pdf?refreqid=excelsior%3Ac4a1186a80f06e4a38bdca8fb50df792)
See Figure in Brian Taylor (The ants of Africa; Chapter 3) illustrating directly the importance of shade trees in cocoa plantations: http://antsofafrica.org/ant_species_2012/mosaics9.htm

Results
Daytime. The authors need to keep in mind that fogging and baiting are two complementary techniques. It is therefore normal that differences occurred. As said above, the fogging gave an ‘instantaneous image’ of what ant species were active during the short period of time of the action of the insecticide, whereas baiting showed what ant species were exploiting the bait when the bait was verified by the authors. So, if the period of time is short, the bait permits researchers to gather ‘opportunist species’ whose workers quickly locate recently placed baits and recruit nestmates. They are generally very timid and abandon the baits when workers of more aggressive species, or extirpators, arrive. Combats can occur between successive extirpator species (this is described in Hölldobler & Wilson 1990, cited in the text).
Thus, placing baits once during daytime and once at night is interesting but the information provided is relatively limited as it likely captures different scenarios. The whole, in addition to fogging, gives a relatively good idea of the ant species diversity, but less of the ant mosaic itself. Yet, through the statistical approach, the authors, interestingly, show that some species are numerically dominant in certain situations, but not in the others (particularly, the difference between daytime and night using baits). Due to the peculiarity of Dinomyrmex gigas, I suggest that the authors develop more the fact that this species was present at night on 14 trees out of 20 (see lines 194-195 and then 198-200).

Lines 195-201. It is frequent that tree crowns are shared by a diurnal and a nocturnal species, both being dominant, but with different rhythms of activity (they were called ‘co-dominant species’ by Majer 1976; see also more citations in Dejean et al. 2007 cited in your manuscript). I therefore suggest deleting lines 196-197 ‘a lower degree of turnover than that found between 197 baiting and fogging during the day’, something that corresponds to two different approaches.

Discussion
I suggest modifying or deleting the first paragraph of the Discusion (lines 212-218) as it seems to be a kind of mea culpa by Tom Fayle, whereas lines 260-263 explain very, very well the situation. Furthermore, the authors need to remember that they only indirectly worked on an ant mosaic. Better to explain why the results obtained can be correlated to the notion of ‘ant mosaics’ in a short sentence (not obligatory, of course).
Alternatively, the Discussion can begin with the paragraph in lines 219-231. Furthermore, I urge the authors to complete this paragraph with information on Dinomyrmex gigas present at night on 14 out of 20 trees.
Also, because the paper by Davidson et al. (2007) dealing with ants of the rainforest canopy of Brunei is cited, I suggest that the authors write a small paragraph on the strong difference between the dominant ant species they noted and those recorded by Diane Davidson. For example, they did not record Oecophylla smaragdina (a typical territorially dominant arboreal ant species) whereas Diane Davidson did not record Dinomyrmex gigas (but she noted this species during other studies).

Line 232. ‘The robustness of patterns of segregation to the sampling methods employed might be because dominant ants are those that are both highly abundant in general’ for me, it is not ‘might be’ but rather ‘is due to’.

Line 234. As I wrote above, it is normal that the two sampling methods result in different percentages of ant species (or different ant species) as they select for different ant species. So, better to explain that difference in the Materials and Methods and here to comment on the complementarity of the two methods.

Lines 237-240. There is ONE ant mosaic in the canopy of the forest, not in one isolated tree (i.e. one tree is only part of the mosaic if the ant mosaic exists, something needing to be proven). I suggest that you delete this paragraph rather than to bother writing a new one.

Line 240-243. It is normal that no ant species totally dominates the tree crowns (i.e, not the canopy; you did not work on it) as you gathered both dominant species and the non-dominant they tolerate. So, better to delete this part and to adapt the following paragraph on Vollenhovia (nota: I hope that you sent specimens to well known taxonomists such as Barry Bolton and Brian Fisher as it remains possible that this myrmicine species is a territorially dominant arboreal species with large polydomous colonies).

Lines 252-265. I suggest that you simplify this paragraph in lines 260-263 whose meaning is clear. Maybe, it can be placed elsewhere in the text (so permitting to delete the rest). Anyway, lines 264-265 can be questioned as Dinomyrmex gigas workers climb up the trees each night to attend hemipterans in the crown.

Lines 269-271. It is easy to verify in your data if the dominant species differed between trees, so that I suggest deleting ‘might be’ by ‘was’ after verification.

Lines 271-274. Better to delete this paragraph as differences between methods during the daytime will be already explained in your next version of the manuscript, whereas differences in baiting between day and night are due to different rhythms of activity. The same for lines 275-279, even if temperature also plays a role (note that certain arboreal ants are active at night only during a few hours). Your work is interesting, this is why I suggest to avoid speculations and ways to shorten the text.

Lines 280-286 also can be deleted or adapted.

Line 288 needs to be modified. Yours results indicate that dominant ants are present in the emergent trees, and so are in line with the notion of ant mosaic that is likely present in the rest of the rainforest canopy.

References
To simplify, I grouped the lines needing the same correction.
1- Genus and species names in italic letters; lines: 299, 380, 389; 392; 399-400.
2- Many letters in uppercases need to be written in lowercase; lines: 335-336; 349-350; 382.
3- Line 322: delete one of the two dots before Insect Science.
4- Line 327: Djiéto (not Djieto).
5- Line 337: Abhandlungen der Senckenbergischen Naturforschenden Gesellschaft, Stuttgart.
6- Line 370. I hope that you accessed this site after 2013, so, five years ago.
7- Lines 386-388. I suggest deleting this reference, see above.

Figures
Do not forget to modify Fig. 2 to permit readers to separate Dinomyrmex gigas from the four other ant species appearing in black.

Conclusion of the review
This new version of the manuscript shows major improvement over the previous version. In my opinion, if the authors coordinate themselves well and because we now live in the “computer era”, the changes still needed can be made in less than two weeks, so that I recommend acceptance with “minor corrections”. I agree to review the next version if the associate editor feels that this is necessary.

·

Basic reporting

All comments by the two referees have been adequately attended to.

Experimental design

All comments by the two referees have been adequately attended to.

Validity of the findings

All comments by the two referees have been adequately attended to.

Comments for the author

The manuscript has now been substantially improved and concerns of both referees have been adequately addressed. I have a few editorial comments to add:

Line 34 ‘in a new forest statum’ – a better wording might be ‘in a hitherto under-investigated’ or simply ‘in an under-investigated’. The way it is currently worded implies that this stratum has just developed.

Line 85 I would say ‘’those with generally less populous colonies’ as there could by minute, non-dominant species with large numbers of ants in the colony.

Line 125 Suggest adding ‘and’ before ‘leading’

Line 238 Throughout, I suggest hyphenating ‘within-tree’ and between-tree’. You have done this on line 242.

Line 347 Suggest deleting the word ‘average’ before ‘bait’

Line 357 Don’t you mean ‘may not be valid’?

Line 363 ‘result’ , not ‘resulting’

Line 498 Remove caps in title of reference

When these are attended to I believe that the article should be ready for publication.

---

## Round 0.3 · accepted · Accept

Thanks for the careful and well laid out responses to reviewers. Congrats